# Long-baseline quantum sensor network as dark matter haloscope

Min Jiang[1,2,3,12], Taizhou Hong[1,2,3,12], Dongdong Hu[3,4,12], Yifan Chen[5], Fengwei Yang [6], Tao Hu[7], Xiaodong Yang[7], Jing Shu[8,9,10] ✉, Yue Zhao [6] ✉, Xinhua Peng [1,2,3] ✉ & Jiangfeng Du [1,2,3,11]

Ultralight dark photons constitute a well-motivated candidate for dark matter. A coherent electromagnetic wave is expected to be induced by dark photons when coupled with Standard-Model photons through kinetic mixing mechanism, and should be spatially correlated within the de Broglie wavelength of dark photons. Here we report the first search for correlated dark-photon signals using a long-baseline network of 15 atomic magnetometers, which are situated in two separated meter-scale shield rooms with a distance of about 1700 km. Both the network's multiple sensors and the shields large size significantly enhance the expected dark-photon electromagnetic signals, and long-baseline measurements confidently reduce many local noise sources. Using this network, we constrain the kinetic mixing coefficient of dark photon dark matter over the mass range 4.1 feV-2.1 peV, which represents the most stringent constraints derived from any terrestrial experiments operating over the aforementioned mass range. Our prospect indicates that future data releases may go beyond the astrophysical constraints from the cosmic microwave background and the plasma heating.

Despite the astrophysical evidence for the existence of dark matter over the past eight decades, direct detection of its non-gravitational interactions with Standard-Model particles and fields remains elusive. To address this, numerous theories have been proposed, many of which posit the existence of new fundamental particles beyond the Standard Model that could make up dark matter. Among these candidates, a class of ultralight bosons, such as axions[1–3] and dark photons[4], stand out as particularly well-motivated. The existence of these bosons is naturally predicted within fundamental theories positing extra dimensions[5–8]. When their mass is below $\mathcal{O}(1)$ eV, they behave like coherent waves with a large occupation number within a given correlation length and time.

Numerous experimental efforts have been undertaken to detect ultralight bosonic dark matter. To date, previous searches have primarily focused on axion and axion-like particles employing inverse Primakoff effects, where they convert into photons in a strong magnetic field background[9–12]. Examples of such experiments include ADMX[13], CAPP[14], HAYSTAC[15], ORGAN[16], DM Radio[17], and

[1]CAS Key Laboratory of Microscale Magnetic Resonance and School of Physical Sciences, University of Science and Technology of China, Hefei, Anhui 230026, China. [2]CAS Center for Excellence in Quantum Information and Quantum Physics, University of Science and Technology of China, Hefei, Anhui 230026, China. [3]Hefei National Laboratory, University of Science and Technology of China, Hefei 230088, China. [4]State Key Laboratory of Particle Detection and Electronics, University of Science and Technology of China, Hefei, Anhui 230026, China. [5]Niels Bohr International Academy, Niels Bohr Institute, Blegdamsvej 17, Copenhagen 2100, Denmark. [6]Department of Physics and Astronomy, University of Utah, Salt Lake City, UT 84112, USA. [7]Suzhou Institute of Biomedical Engineering and Technology Chinese Academy of Sciences, Suzhou, Jiangsu 215163, China. [8]School of Physics and State Key Laboratory of Nuclear Physics and Technology, Peking University, Beijing 100871, China. [9]Center for High Energy Physics, Peking University, Beijing 100871, China. [10]Beijing Laser Acceleration Innovation Center, Huairou, Beijing 101400, China. [11]Institute of Quantum Sensing and School of Physics, Zhejiang University, Hangzhou 310027, China. [12]These authors contributed equally: Min Jiang, Taizhou Hong, Dongdong Hu. ✉e-mail: jshu@pku.edu.cn; zhaoyue@physics.utah.edu; xhpeng@ustc.edu.cn

ABRACADABRA[18]. On the other hand, laboratory searches for kinetically mixed dark photon dark matter (DPDM) do not depend on electromagnetic background fields. DPDM can generate resonant cavity modes or magnetic fields in an electromagnetic shielded room through effective oscillating currents[19]. Previous constraints on axion–photon coupling have been reinterpreted into those for DPDM due to their similarities in detection mechanisms[20]. Recently, DPDM has also been searched for using various strategies, including resonant LC circuits[19], dish antennas[21], geomagnetic fields[22], atomic spectroscopy[23], and radio telescopes[24]. Certain experiments, such as FUNK[25], SuperMAG[26], and QUALIPHIDE[27], have already set experimental constraints on DPDM. Nevertheless, DPDM mass constraints below neV mostly rely on astrophysical and cosmological observations, such as anomalous heating up of the plasma[28–31] and distortion of the cosmic microwave background (CMB)[31,32], both of which are dependent on astrophysical modeling.

In this article, we demonstrate the first Search for Dark Photons with synchronized Atomic Magnetometer Arrays in Large Shields (AMAILS), which consist of 15 atomic magnetometers. Such magnetometers are situated in two separate electromagnetic shielded rooms in Harbin and Suzhou, China, with a distance of about 1700 km between the locations (Fig. 1a), and are synchronized with the global positioning system (GPS). These magnetometers exhibit an exceptionally high level of sensitivity at the femtotesla level and offer the capability of measuring the magnetic-field radio induced from the DPDM in proximity to the wall of the shield room. By correlating the readouts of separated magnetometers in the network, we report the first long-baseline network of quantum sensors searching for DPDM-correlated signals over 1000 km. Our experiments constrain the parameter space describing the kinetic mixing of dark photons over the mass range from 4.1 feV to 2.1 peV, which exceeds that of state-of-the-art terrestrial DPDM searches[33–35]. We would like to emphasize the difference in DM search approaches between this work and other studies. Previous search experiments on wave-like DM have mostly been limited to searches for local signals with single detector[17–19], where it is challenging to confidently distinguish DM signals from local technical noises. Recently, the use of networked quantum sensors for DM searches has been proposed[36–38] or demonstrated, including a global network of optical magnetometers (GNOME)[39] and networks of atomic and optical clocks[40,41]. Such networks consider the situation in which dark-matter particles interact with each other and generate topological defects instead of wave-like DM. Previous work[26] has employed the SuperMAG network, initially intended for measuring Earth's magnetic field with classical magnetometers, to explore DPDM. In contrast, our research employs quantum magnetometers specialized for dark matter investigations, providing us with enhanced potential for improvement and the incorporation of novel features within a distinct mass range. Although the GNOME network has the potential to be applied to wave-like DPDM searches, their optical magnetometers are placed in the center of the small-scale shielded rooms, typically with an innermost diameter on the order of 10 cm. In this case, the DPDM signal is about three orders of magnitude smaller than that in our experiments (Supplementary Section I, Fig. S1). Unlike previous works, our work opens intriguing opportunities to search for wave-like DM with long-baseline quantum sensors, and with further optimization, it is promising to reach an unexplored parameter space beyond the astrophysical constraints imposed by the anomalous plasma heating[28–31] and CMB distortion[31].

## Results

The dark photon is a new massive U(1) gauge boson that kinetically mixes with the electromagnetic photon through an interaction term $\varepsilon F_{\mu\nu} F'^{\mu\nu}/2$ [19,24]. Here $F^{\mu\nu}$, $F'^{\mu\nu}$ represent the field strength tensor of electromagnetism and the dark photon, respectively, and $\varepsilon$ denotes the kinetic mixing coefficient between the two. When the mass of dark photon $m_{A'}$ is below $\mathcal{O}(1)$ eV, the dark photon behaves as a coherent wave with a frequency nearly equal to $m_{A'}$. Due to the mixing of the dark photon and electromagnetic photon, the dark photon can induce harmonic oscillations of electrons inside the wall of the electromagnetic shied room (Fig. 1b). As a consequence, this leads to a monochromatic radio signal that can be described with an effective current $\vec{J}_{\text{eff}}$. Specifically, in the interaction basis, the dark photon produces an effective elctromagnetic current $J^\mu_{\text{eff}} = -\varepsilon m_A^2 A'^\mu$ [19], where $A'_\mu$ is the gauge potential of dark photon. The current strength can be estimated by the Galactic dark-matter energy density $\rho \approx m_{A'}^2 |\vec{A}'|^2/2 \approx 0.45$ GeV/cm³ [1–3]. Details of dark photon electrodynamics are present in Supplementary Section I.

In order to detect the electromagnetic signal produced by the effective current, we use meter-scale electromagnetic shields capable of converting magnetic fields within their confines. Our experimental setup comprises a network of synchronized atomic magnetometers that are operated within two separate shield rooms - one located in Harbin and the other in Suzhou—with a distance of 1692 km between them (Fig. 1a). Both rooms are made of five-layer mu-metal, and their innermost layer is cuboid in shape with the dimension of $2 \times 2 \times 2$ m³. The magnetometer captures DPDM-induced $\vec{J}_{\text{eff}}$ along the $z$-axis, which subsequently produces a radio B-field that runs parallel to the walls in a horizontal tangential direction. Please refer to Fig. 1b for a visual depiction of this phenomenon, which is also discussed in Supplementary Section I. We find that the field signal reaches a maximum on the surface of the wall. The amplitude of such an oscillating magnetic field in the vicinity of the shied wall is approximated as

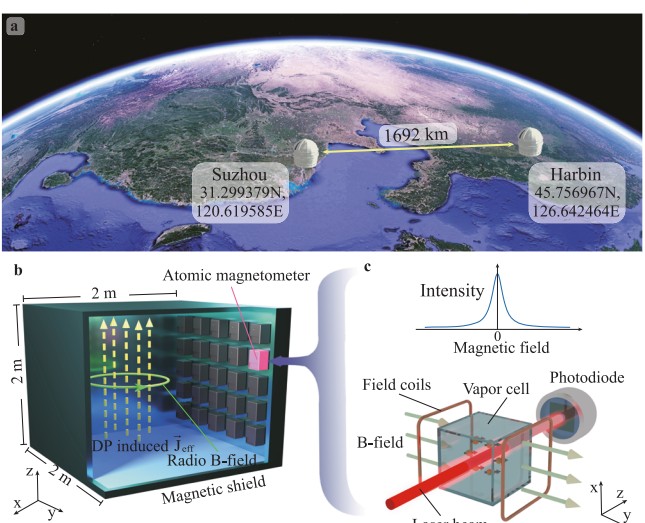

**Fig. 1 | Quantum sensor network. a** A network of 15 atomic magnetometers situated in two separate shield rooms in Suzhou and Harbin, China, with a distance of 1692 km between them. Such magnetometers are synchronized with the global positioning system. The underlying map is from Google Earth. **b** Schematic for dark photon dark matter (DPDM) induced radio signal inside a shielded room. Each room is made of five-layer mu-metal and its innermost layer has the dimension of $2 \times 2 \times 2$ m³. A magnetic field can be produced by the DP radio current $\vec{J}_{\text{eff}}$ (dashed line) along $z$. Miniaturized atomic magnetometers (QuSpin Inc.) are installed on the surface of the shield room and detect the DP induced magnetic field (green circle) along the $y$ direction parallel to the wall. **c** Schematic illustration of atomic magnetometer based on zero-field resonance. Top, operation in zero field; bottom, operation with a measured field that causes alkali-metal spin precession and reduces the transmitted light intensity.

$$B \approx |\vec{J}_{\text{eff}}| V^{1/3} \approx 1.63 \times 10^{-12} \, \varepsilon \left(\frac{m_{A'}}{10 \, \text{Hz}}\right) \left(\frac{V^{1/3}}{1 \, \text{m}}\right) \text{T}, \tag{1}$$

and becomes smaller when approaching the room center. Importantly, the strength of the magnetic field induced by DPDM is directly proportional to the length scale $V^{1/3}$ of the shield room. This implies that a larger shield room, analogous to the function of a radar collecting minuscule radio signals, can remarkably enhance the search for DPDM.

We employ atomic magnetometers as DPDM detectors. Atomic magnetometers are recognized as a type of quantum sensor[42] that exploits the quantum phenomenon known as the spin-exchange relaxation-free (SERF) effect[43,44], thereby enhancing atomic coherence time and improving measurement sensitivity. As shown in Fig. 1c, each magnetometer we used comprises a vapor cell that contains a droplet of isotopically enriched $^{87}Rb$, weighing several milligrams and approximately 700 torr buffer-gas $N_2$. To optically polarize $^{87}Rb$ atoms, we utilize a circularly polarized laser beam with its frequency tuned to the center of the buffer-gas broadened and shifted D1 line of $^{87}Rb$. In the absence of a magnetic field, the spin magnetic moments align with the pump beam, resulting in the maximization of laser light transmission to the photodiode. However, a magnetic field perpendicular to the beam causes Larmor precession, which rotates the magnetic moments away from alignment and, consequently, leads to a detectable decrease in light transmission. This effect produces a zero-field resonance that acts as a highly sensitive magnetic field indicator. Our network utilizes miniaturized SERF magnetometers made by QuSpin Inc. that are compacted to a size of just a few centimeters and achieve a magnetic field sensitivity of approximately 15 fT/Hz$^{1/2}$.

At the heart of our approach lies an array of atomic magnetometers that are installed in two separate observation stations. Based on the DP mass range in our search, the correlation length of DPDM ($\approx 10^{-6}$ eV/$m_{A'}$ km) is larger than the baseline (~1700 km) between the aforementioned two stations. As a result, DPDM would produce a correlated common-mode magnetic field in all spatially separated sensors. While a single atomic magnetometer could in principle, detect the radio signal from DPDM, under realistic experimental conditions, it would be challenging to confidently distinguish the DPDM signal from many sources of noise. To tackle this problem, we simultaneously monitor multiple magnetometers, with 13 magnetometers installed in Suzhou station and 2 magnetometers in Harbin station, in order to extract potential events from their correlated signals. The magnetic field data from these magnetometers are recorded using custom data-acquisition systems that are synchronized to GPS time. We calculate the cross-correlation spectra for every magnetometer pair shown in Fig. 2a, where the cross-correlations between magnetometers in the same shield room or in separate shield rooms are calculated. For example, Fig. 2b displays the cross-correlation spectrum of two magnetometers, where some sharp technical noise appears in only one of the sensors and thus can be easily distinguished. The details of the quantum sensor network are present in Supplementary Section III.

The sensitivity of our search for dark photon dark matter increases with the number of quantum sensors in the network. There are $C_{15}^2 = 105$ correlators in total, which enable us to obtain the cross-correlation spectrum for each pair of synchronized magnetometers. We depict the average spectrum of all cross-correlations as a function of the correlator number $\mathcal{N}$ (Fig. 2c). The results demonstrate a significant improvement in magnetic field sensitivity at different frequencies, namely, increasing from approximately 15 fT/Hz$^{1/2}$ to almost a few fT/Hz$^{1/2}$. To clearly illustrate this, we plot the sensitivity at 10.1 Hz as a function of the correlator number $\mathcal{N}$ in Fig. 2d. The corresponding sensitivity data can be well fitted with the function of $\mathcal{N}^{-a}$, where $a \approx 0.25$. Finally, the network sensitivity approaches approximately 4.2 fT/Hz$^{1/2}$ at 10.1 Hz. As shown in Fig. 2c, this level of network sensitivity is attained at most other frequencies.

We further analyze the signal-to-noise ratio (SNR) distribution for all data points in the real part of the cross-correlation spectrum ranging from 1–500 Hz (Supplementary Section IV). Specifically, we plot

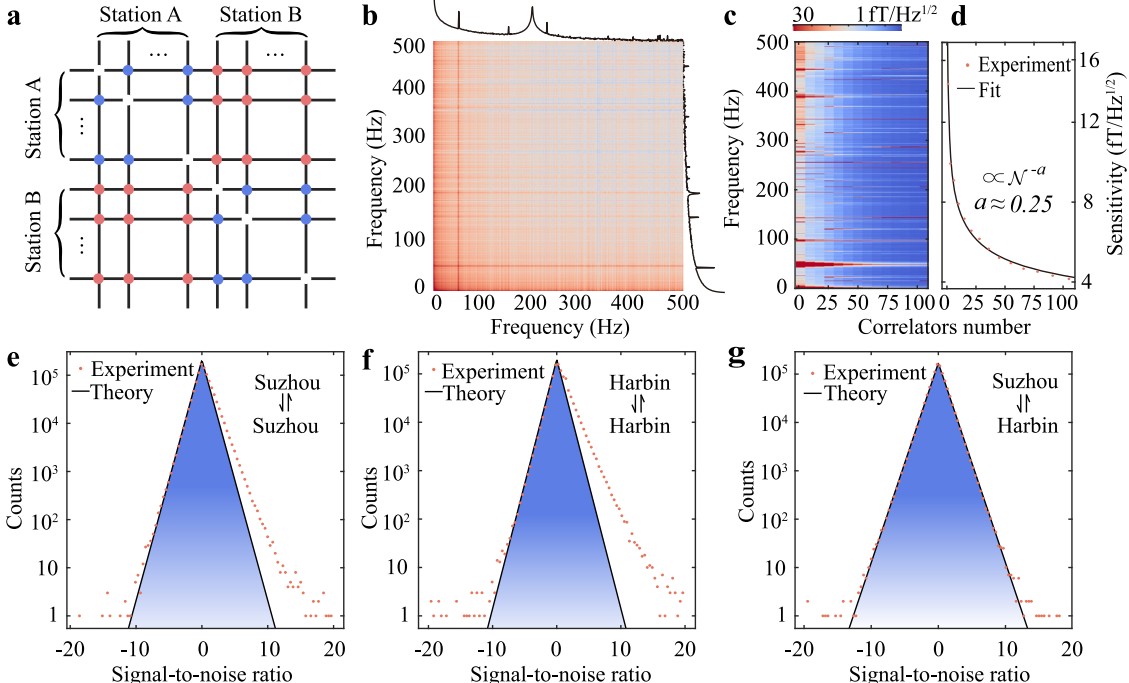

**Fig. 2 | Cross-correlation of sensor network. a** Diagram of the cross-correlation between the atomic magnetometers located in station A (Suzhou) and station B (Harbin). The blue dots and red dots represent the cross-correlation from the magnetometers in the same station and different stations, respectively. **b** Two-dimensional cross-correlation spectrum of two sensors. **c, d** The network sensitivity as a function of correlators number $\mathcal{N}$. As an example, The magnetic sensitivity of network at 10.1 Hz increases with increasing correlators number and can be fitted with the function of $\mathcal{N}^{-a}$ ($a \approx 0.25$). Distribution of the signal-to-noise ratio (SNR) of the cross-correlation spectrum between two sensors: **e** two sensors in Suzhou station; **f** two sensors in Harbin; **g** one sensor in Suzhou station and the other in Harbin station. The SNR distributions in **e** and **f** are asymmetrical due to the common-mode magnetic noise ($\approx$5 fT/Hz$^{1/2}$) of the shield room that only contributes to the positive part of the SNR. In contrast, the SNR distribution of two sensors from different stations is symmetrical without common-mode noise.

the SNR distribution for two magnetometers located in the same shield room, for example, at Suzhou station (Fig. 2e) or Harbin station (Fig. 2f). Both show an unexpected asymmetric distribution, with the right side slightly exceeding the left, deviating from the theoretical prediction (see the blue shaded regions). By performing numerical simulation, we reveal that this occurs due to the common-mode magnetic field of the shield room (Supplementary Section V). The common-mode noise is approximately estimated to 5 fT/Hz$^{1/2}$, which agrees with the direct calculation of the magnetic noise within the shield room. In contrast, the SNR distribution of the long-baseline cross correlation between magnetometers in Harbin and Suzhou is symmetric (Fig. 2g) because their noise is uncorrelated between two separate shields. This demonstrates the unique ability to distinguish common-mode noise with distributed sensors. For instance, there are 21628 data points that lie outside the theoretical prediction region in Fig. 2e. In contrast, in Fig. 2g, there are only 1349 points in outside the region, indicating a significant 16-fold noise suppression. Based on the results of the numerical simulation, the dominant common-mode noise is due to the shield room. However, there exist large-scale noises potentially impacting the experiment, such as the solar wind, geomagnetic storm, and Schumann resonances, and separating these two sites at a long distance can effectively reduce these noises.

Having established the network technique, we perform the broadband search for dark photon dark matter in the frequency range from 1 to 500 Hz, corresponding to dark photon masses ranging from 4.1 feV to 2.1 peV. Throughout the experiment, we ensure that all quantum sensors are accurately synchronized with GPS time and record 2000 seconds of signal data; further, we calculate all pairs of cross-correlation spectra and their average spectrum and produce the SNR distribution of the average spectrum, present in Fig. 3a. In order to determine the detection threshold, we carry out Monte Carlo simulation to estimate the 95% confidence level (C.L.) for SNR linked to the measured kinetic mixing coefficient $\varepsilon$ (Fig. 3a). Our analysis procedure yields 318863 potential DPDM candidates that exceed the 95% C.L. We validate the possibility of true DPDM signals and exclude all candidates. We check the viability of our data analysis procedure by inserting simulated DPDM signals into our data and confirm that the analysis method can recover

these signals with their correct kinetic mixing coefficients. The detailed procedures of data processing, exclusion, and testing are presented in Supplementary Sections VI and VII.

Figure 3b presents our new constraints on the kinetic mixing coefficient $\varepsilon$ with 95% C.L. The present work explores a mass range from 4.1 feV to 2.1 peV, overlapping with the Earth[33] and Jupiter[34] experiments and Cavendish-Coulomb experiment[45] and substantially improves previous limits. For example, our experiment places a constraint $\varepsilon \approx 5 \times 10^{-6}$ at 2.1 peV, surpassing the previous limits from Cavendish–Coulomb experiments by about three orders of magnitude. We also show the comparison with astrophysical constraints in Fig. 3b, for example, the cosmic microwave background photon's transition to dark photon[31,32], the DM-induced heating on the plasma[29], the intergalactic medium[28], and the dwarf galaxy Leo T[30]. These constraints strongly rely on the cooling rates of these systems and the understanding of their cooling mechanisms. Furthermore, we note that stringent limits are obtained by imposing an upper limit on the energy deposition during the epoch of He-II reionization through Lyman-$\alpha$ observations[31,32]. However, these constraints are subject to the influence of significant astrophysical uncertainties, resulting in varying results depending on several key factors: (i) the probability density functions of plasma mass at early times, particularly pertaining to the tail distribution; (ii) the maximum cutoff on density perturbation scales; and (iii) the transportation of energy injection, which relates to the selection of sensitivity functions and thresholds. Given the aforementioned uncertainties, it is important to conduct a terrestrial experiment to investigate the comparable mass range.

## Discussion

A further improvement in experimental sensitivity to dark photon dark matter is anticipated. Increasing the number of spatially separated magnetometers up to 1000 could improve the DPDM search sensitivity by a factor of about 10; another potential factor to consider is the length scale of the shield room, which can be expanded up to 20 m in size to maximize the potential dark photon signal. In fact, numerous large-scale mu-metal shielded rooms already exist in the fields of magnetoencephalography for the brain and magnetocardiography for

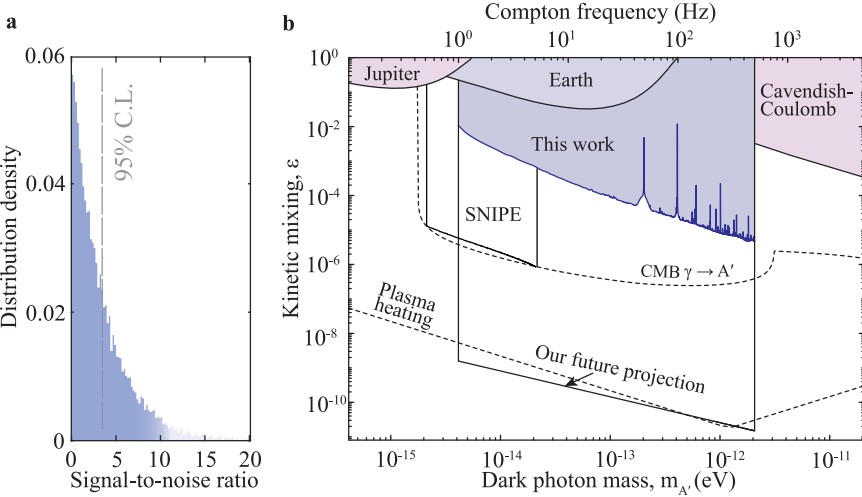

**Fig. 3 | Results of dark photon dark matter search. a** Distribution of the signal-to-noise ratio of the network-averaged cross-correlation spectrum. The 95% confidence level (C.L.) is determined with Monte Carlo simulation. **b** Limits on kinetic mixing $\varepsilon$ of dark photon dark matter in the mass range from 4.1 feV to 2.1 peV. The blue-shaded region is excluded from our network measurement at 95% C.L. The black line shows our future projection of an upgraded magnetometer network. The light-blue shaded regions show other constraints derived from terrestrial or extra-

terrestrial experiments, including observing the magnetic fields of Earth[33] and Jupiter[34], measurements with a network of magnetometers in quiet magnetic environments (SNIPE)[59], and Cavendish-Coulomb experiments[35]. The dashed lines show the limits from cosmological and astrophysical observations, including cosmic microwave background (CMB) photon's transition to dark photon[31,32], and DPDM heating the plasma[28–30].

the heart[46,47]. These rooms possess over 1000 miniaturized magnetometers in aggregate and could potentially be repurposed during their unoccupied periods to create a DPDM search network. Moreover, an additional efficient approach would involve optimizing the magnetic sensitivity of atomic magnetometers, which are currently considerably far from reaching their standard quantum limits. With the development of quantum technology, it could potentially approach the standard quantum limit of $0.01 \, fT/Hz^{1/2}$. To enhance sensitivity even further, a promising strategy involves the installation of magnetometers at various positions or orientations within each shielded room. The information collected by these magnetometers can then be combined in an array to effectively reduce local common-mode noise (see Supplementary Information Section I). By utilizing these promising magnetometers and investing in one year of integration, we can significantly enhance our search sensitivity, attaining an improvement of up to 6 orders of magnitude beyond our present limits, as depicted by the solid line in Fig. 3b. This would open avenues to explore unexplored parameter space beyond the astrophysical constraints imposed by CMB distortion[31,32] and plasma heating[28–30].

Including other types of magnetometers can lead to the extension of dark-photon search masses. As demonstrated in Ref. 48, radio-frequency atomic magnetometers have exhibited a level of sensitivity below $1 \, fT/Hz^{1/2}$ and can search for dark photons at a mass range of 10 kHz to 10 MHz. For example, recent studies have demonstrated a noise floor of $0.5 \, fT/Hz^{1/2}$ at 300 kHz[48], resulting in a dark-photon search sensitivity of $\epsilon \approx 10^{-10}$ for 100 s of integration. This notable sensitivity is at least three orders of magnitude better than the constraints from the Cosmic Microwave Background[31,32]. Promising sensors for dark photon detection in the higher frequency regime can be realized using nitrogen-vacancy center-based magnetometers that have recently shown ~10 $pT/Hz^{1/2}$ sensitivity in GHz microwave detection[49,50]. For uncovering lower masses within the range of mHz to Hz, one can use noble-gas spin masers[51,52] that have high sensitivity in the Hz range or install magnetometers on a stable rotating table, which up-converts the low-frequency DPDM-induced field to a higher frequency.

In summary, we demonstrate a network of synchronized atomic magnetometers in large shields to search for dark photon dark matter. Here, we assume a standard halo model for the momentum distribution of DPDM that forms after virialization. However, it is essential to note that other types of dark matter distributions, such as anisotropic ones, like a local cold stream due to the merger of the galaxy, have been considered[53–55]. Moreover, dark photon fluxes from black holes[56] or other cosmological sources may exist without being the dominant form of dark matter. Our network has the potential to play an important role in determining the nature of these sources[38,57]. Excellent localization is expected through the use of directional detection inside each shield room and the long baselines that separate various shield rooms. Consequently, such a network of synchronized quantum sensors can associate multi-messenger astronomy observations together with gravitational wave detectors and electromagnetic observatories[58].

## Data availability
The source data and code that support the plots in this paper are available from the corresponding author upon request.

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

## Acknowledgements

This work was supported by the Innovation Program for Quantum Science and Technology (Grant No. 2021ZD0303205), National Natural Science Foundation of China (grants Nos. T2388102, 11927811, 12150014, 12205296, 12274395), Youth Innovation Promotion Association (grant No. 2023474), and Chinese Academy of Sciences Magnetic Resonance Technology Alliance Research Instrument and Equipment Development/Functional Development (Grant. No. 2022GZL003). Y.C. is supported by VILLUM FONDEN (grant No. 37766), by the Danish Research Foundation, and under the European Union's H2020 ERC Advanced Grant "Black holes: gravitational engines of discovery" grant agreement no. Gravitas-101052587, and by FCT (Fundação para a Ciência e Tecnologia I.P, Portugal) under project No. 2022.01324.PTDC. J.S. is supported by Peking University under startup Grant No. 7101302974 and the National Natural Science Foundation of China under Grant Nos. 12025507 and 12150015 and is supported by the Key Research Program of Frontier Science of the Chinese Academy of Sciences (CAS) under Grants No. ZDBS-LY-7003 and CAS project for Young Scientists in Basic Research YSBR-006. F.W.Y. and Y.Z. are supported by the U.S. Department of Energy under Award No. DESC0009959. *Note*: After completion of this work, we became aware of recent work on the search for hidden photons and axion dark matter. In Ref. 45, this work excludes dark photons in the frequency range from 0.5 to 5 Hz with a constraint of $1 \times 10^{-5}$ to $1 \times 10^{-6}$ on the kinetic mixing parameter.

## Author contributions

M.J. designed experimental protocols, analyzed the data, and wrote the paper. T.Z.H. and D.D.H. performed search experiments, analyzed the data, and wrote the paper. Y.F.C. and F.W.Y. analyzed the data and wrote the paper. T.H. and X.D.Y. performed search experiments. J.S., Y.Z., and X.H.P. proposed the experimental concept, devised the experimental protocols, and edited the paper. J.F.D. contributed to the design of the experiment and proofread and edited the paper. All authors contributed with discussions and checking the paper.

## Competing interests

The authors declare no competing interests.
