## [Peer Review File · Nature Communications]

Long-Baseline Quantum Sensor Network as Dark Matter HaloscopeReviewer #1 (Remarks to the Author):

The manuscript titled "Long-Baseline Quantum Sensor Network as Dark Matter Haloscope" by Jiang M. et al presents a dark matter search for dark photons.

The experiment presented consists of using large ($\sim 2 \times 2 \times 2$ m) magnetic shields that would transduce a dark photon field into a small magnetic field, that was searched for using an array of atomic magnetometers.

By correlating the signals of the magnetometer arrays in Suzhou (13 magnetometers) with those in Harbin (2 magnetometers)--which are separated by ~ 1700 km--they can suppress the effects of local common-mode noise.

Their work presents the first laboratory limits in new explored parameter space for dark photons. Additionally, they implement an (arguably) new experimental scheme, and therefore it is of interest to the community dedicated to search for ultra-light bosonic dark matter; and dark matter in general.

The work in the main manuscript is well-written and presented in a clear manner that can be easily understood. Additional important details are also given in supplemental material that go in further depth in the analysis techniques, model and assumptions.

The originality is a bit overinflated in the text, as:

1.- The work is very similar to the idea presented in (Derevianko's 'Detecting dark-matter waves with a network of precision-measurement tools' PRA 97, 042506 (2018)), which they reference [Ref. 37].

2.- I believe networks of sensors have been used previously to search for wave-like dark photon signatures in SUPERMAG (which they reference).

3.- They also adequately reference GNOME, an experiment that has obvious parallels, as both are networks of atomic magnetometers inside magnetic shields for DM searches.

Nonetheless, the claim that they are the first network of 'quantum' sensors looking for wave-like dark photon DM is probably correct.

In my view, they should reference another experiment which gives important context, as it also excludes dark photons in their frequency range:

by I. A. Sulai et al "A Hunt for Magnetic Signatures of Hidden-Photon and Axion Dark Matter in the Wilderness" (<https://arxiv.org/abs/2306.11575>)

Which excludes dark photons in the 0.5 - 5 Hz range with a sensitivity of $1e-5$ to $1e-6$ in the kinetic mixing parameter (and their limits should be included in fig. 3b in my opinion).

Broadly speaking, the methodology is sound and there are enough details (if one includes the supplementary material) to reproduce these results.

However, there are a few points that I would like to raise:

For their correlations, they take data for 2000s at a sampling rate of 1000Hz -> so their single sided spectra should have 1M points.

However, their dark matter candidate bins are $\sim 320k$, which is much more than one would expect, and a significant fraction of the spectra (keep in mind this is at the 95% confidence limit!).

To me, this suggests that the "detection threshold" is (and finally their exclusion limits) being set too low; looking through the supplementary material, I think they are missing a statistical correction that should be dependent on the total number of frequency bins.

I find the idea of introducing a lag into one of the channels in order to suppress the DM signal and determine the background, clever.

If I understand correctly; they went through all their DM candidate bins and made sure they were consistent with the background using this technique.

What worries me is that the limits presented in fig. 3b are the "detection threshold", when the background is higher than the "detection threshold" (and therefore limits can not be placed at the threshold level)- is this the case?

ie. if a frequency bin has a detection threshold of SNR 2.5; and the measured bin SNR is 6; it will be

flagged as a candidate.- after it is confirmed that the background corresponds to "SNR" of 6, is the limit set at 2.5 or 6?

This is particularly relevant given the large fraction/number of candidates.

The authors ignore the width of the DM signal that would arise due to stochastic fluctuations (G.P. Centers et al, Nature Communications volume 12, Article number: 7321 (2021)) in their analysis. The argument given is that the width (1/correlation time) at their maximum mass of 500 Hz matches their bin width of 0.5 mHz.

Nonetheless, scalloping and other windowing effects can lead to a significant reduction in the bin power (see for example Spectrum and spectral density estimation by the Discrete Fourier transform (DFT), including a comprehensive list of window functions and some new flat-top windows (2002) by G. Heinzel et al)

This effect would only be relevant for the higher frequencies in their limits. I would suggest either correcting the limits to include this or explicitly neglecting stochastic effects.

The authors also bring up the spatial dependence (similar to an TM mode) of the field that would be generated by the dark photons.

Maybe a way to further eliminate local common mode noise and boost sensitivity would be to combine the information of each magnetometer in the array, using their positions and orientation in this TM mode, before the correlation.

Reviewer #2 (Remarks to the Author):

The authors report new dark photon dark matter search limits 15 high-sensitivity magnetometers. The hunt for dark matter is one of the leading puzzles in elementary particle physics, and it has a profound transformative impact once discovered. It is extremely challenging and, hence, highly rewarding to make new progress in the field. Particularly impressive, the manuscript reports world-leading laboratory-based sensitivities in the 10-500 Hz regime and has a future projection that is possible to exceed the best astrophysical limits. The research is of high importance and high quality. The 15 sensors are distributed in two sites, with 13 in Harbin and 2 in Suzhou. The cross-correlations between them would be a signature of the signal, which the authors use to suppress background.

I encourage the authors to address the following issues before I can consider recommending for publication:

General:

- (1) In what sense these are quantum sensors? The devices used in this search are far from the standard quantum limit (which arguably have some sense of quantum). If not, the authors should change the name of "quantum sensor network" in the title and the text.
- (2) What's the gain of the long baseline? It seems the authors did not take much advantage of the long baseline, except that the two locations do not share some of the common systematics.

Analysis:

There are several issues with the data analysis, assumptions, and processing:

- (3) What is the purpose of using correlation treatments? It seems there would yield no difference by simply and classically adding all the 15 sensors and performing FT. Why not simply do the classical addition?
- (4) Below Eq. S88, the authors assume a flat noise spectrum in adjacent 50 bins, is such assumption validated by test or other runs?
- (5) The calculation of SNR and the corresponding signal limit setting process matters a lot for this works. In supplemental material B, the SNR calculation does show a sizable difference between experimental data and simulation in Fig.4, which does not support the statement "The SNR

distribution within the window of 9.1-10.1 Hz is shown in Fig. S4b, which agrees well with that of the experimental data." Similarly, same sizable differences between the experiment and simulation are shown in Fig.5. These differences should have impacted the choice of SNR=3.49 which translates into 95 C.L. The authors should clarify their treatments.

(6) In the more baseline and standard treatment, the authors found many excess signals. Then the authors used a lagged correction to eliminate false signals. From the description, it is unclear if the authors are lagging in the time-domain (e.g., correlating signals taken at different time, differ by a large period, "we lag one of the signals (i.e., z_n, j) by a large period") or by frequency ("We choose the frequency lag $\Delta j = 10$, which is larger than the intrinsic width of the DPDM signal."). The authors should clarify.

(7) The above treatment, if implemented correctly can help reduce background (false signal) if they are from correlated source, in contrast to the situation where backgrounds are of thermal nature. The authors should comment on the motivation and rigor of the treatment. In particular, the authors used multiple lags whose purpose seems to solely suppress the background, is it legit ("Otherwise, the SNR will be computed again with the lag changed to $\Delta j + 10$ and repeat the above procedure.")? Will it suppress the signal (also see next)?

(8) In S-VI, the authors reconstructed the fake signal but lacked many details. Has the analysis gone through the "lag"-signal elimination process? If so, what is the SNR distribution after this "lag"-signal elimination process? It would also be more systematic that for the same experimental data set, the authors implement several signals at different frequencies and check whether the consistent signal reconstruction is true for different frequencies (so it is robust against background fluctuations).

Others:

(9) (Minor) Given that we do not know the DM direction, we are actually constraining the component of DM along the vertical direction of the experiment. There are a few effects, including the rotation of the direction over 2000 sec of data collecting time, the angular difference between site A and site B, etc. It might be useful to comment on their impact (which is expected to be small but gives an $O(1)$ suppression to the signal rate).

(10) (Minor) Fig.2 should only have two columns and rows for site B, for the configuration realized in this work.

Reply to Reviewer #1

Comment 1-1

The manuscript titled “Long-Baseline Quantum Sensor Network as Dark Matter Haloscope” by Jiang M. et al presents a dark matter search for dark photons.

The experiment presented consists of using large ($\sim 2 \times 2 \times 2$ m) magnetic shields that would transduce a dark photon field into a small magnetic field, that was searched for using an array of atomic magnetometers.

By correlating the signals of the magnetometer arrays in Suzhou (13 magnetometers) with those in Harbin (2 magnetometers)—which are separated by ~ 1700 km—they can suppress the effects of local common-mode noise.

Their work presents the first laboratory limits in new explored parameter space for dark photons. Additionally, they implement an (arguably) new experimental scheme, and therefore it is of interest to the community dedicated to search for ultra-light bosonic dark matter; and dark matter in general.

The work in the main manuscript is well-written and presented in a clear manner that can be easily understood. Additional important details are also given in supplemental material that go in further depth in the analysis techniques, model and assumptions.

Our response: We would like to thank the referee for his/her careful reading and approval of our work. The very insightful and pertinent suggestions really inspired us to improve the quality of our manuscript in many ways. For example, the referee highlighted a crucial stochastic effect that we overlooked in our initial paper. In response, we conducted thorough Monte Carlo simulations and incorporated new discussions on this stochastic effect in our revised paper. Furthermore, we have updated the constraints on our dark-photon dark matter based on these findings. Additionally, the referee suggested a novel approach to enhance sensitivity by combining the information from each magnetometer. We have included one discussion on this suggestion in the revised manuscript. In the following, we provide the detailed reply to each comment and the corresponding changes in the revised paper.

Comment 1-2

The originality is a bit overinflated in the text, as:

1.- The work is very similar to the idea presented in (Derevianko’s ‘Detecting dark-matter waves with a network of precision-measurement tools’ PRA 97, 042506 (2018)), which they reference [Ref. 37].

2.- I believe networks of sensors have been used previously to search for wave-like dark photon signatures in SUPERMAG (which they reference).

3.- They also adequately reference GNOME, an experiment that has obvious parallels, as both are networks of atomic magnetometers inside magnetic shields for DM searches.

Nonetheless, the claim that they are the first network of ‘quantum’ sensors looking for wave-like dark photon DM is probably correct.

Our response: We understand the referee’s concerns and would like to thank the referee for pointing out this comment. These papers are indeed great works in the searches for dark matter and have already been cited in our previous manuscript. In order to clearly claim the originality of our work, we would like to summarize the main differences between our work and these studies.

Firstly, the referenced paper [Derevianko, PRA 97, 042506 (2018)] primarily constitutes a theoretical study centered around the advantages of establishing a network for detecting dark matter. Conversely, our work focuses on developing experimental tools for dark matter searches. Moreover, we employ a novel data analysis technique called “lagged frequency analysis”, which serves as an explicit method to harness the full potential of the network.

Secondly, we agree that our experiments share some similarity with the DPDM search based on SuperMAG data [M.A.Fedderke et al., PRD 104, 095032 (2021)]. However, there are several significant distinctions between our research and the SuperMAG study. The SuperMAG dataset originated from the collaboration focused on geomagnetic observations using magnetometers. These magnetometers rely on classical electromagnetic principles and are limited by classical noise. In contrast, our experiment employs atomic magnetometers specifically designed for dark matter investigations, offering greater potential for sensitivity improvements through additional optimizations. Additionally, the mass range for dark photon in the SuperMAG searches (2×10^{-18} eV to 7×10^{-17} eV) differs from the range examined in our study (4.1×10^{-15} eV to 2.1×10^{-12} eV).

Thirdly, the GNOME network considers the situation in which dark-matter particles interact with each other and generate topological defect instead of wave-like dark matter [S.Afach et al., Nat. Phys. 17, 1396 (2021)]. Although the GNOME network has the potential to search for wave-like dark-photon dark matter, their atomic magnetometers are placed in the center of the small-scale shielded rooms typically with an innermost diameter on the order of 10 cm. In this case, the DPDM signal is about three orders of magnitude smaller than that in our experiments using meter-scale shields.

Considering this, in order to clearly claim our advantages and avoid the potential misunderstanding, we have carefully modified some sentences in the main text. For example,

on page 1, we have modified “...we report the first long-baseline search for DPDM correlated signals...” to “...we report the first long-baseline network of quantum sensors searching for DPDM correlated signals...”,

on page 2, we have added sentences to introduce the SuperMAG work, “Previous work has employed the SuperMAG network, initially intended for measuring Earth’s magnetic field with classical magnetometers, to explore DPDM. In contrast, our research employs quantum magnetometers specialized for dark matter investigations, providing us with enhanced potential for improvement and the incorporation of novel features within a distinct mass range.”

Comment 1-3

In my view, they should reference another experiment which gives important context, as it also excludes dark photons in their frequency range:

by I. A. Sulai et al “A Hunt for Magnetic Signatures of Hidden-Photon and Axion Dark Matter in the Wilderness” (<https://arxiv.org/abs/2306.11575>)

Which excludes dark photons in the 0.5 - 5 Hz range with a sensitivity of $1e-5$ to $1e-6$ in the kinetic mixing parameter (and their limits should be included in fig. 3b in my opinion).

Our response: We appreciate the referee’s help in pointing out this paper for us. This work [“Hunt for Magnetic Signatures of Hidden-Photon and Axion Dark Matter in the Wilderness”, I.A.Sulai et al., Phys. Rev. D 108, 096026 (2023)] is indeed an important experiment searching

for dark photons and is added in the updated reference list. The reason of missing this new paper by I.A. Sulai et al. due to its availability online and submission to arXiv occurring after we completed and submitted our work on May 1st, 2023 (arXiv: 2305.00890). Their paper, on the other hand, became available online and was submitted to arXiv on June 20th, 2023 (arXiv: 2306.11575). We are happy to incorporate their results into our plot and we have included their limits (“SNIPE”) in the updated Fig. 3b. The figure in the revised paper is also shown below. We also added a note at the end of our revised manuscript,

“*Note added. After completion of this work, we became aware of a recent work on search for hidden-photon and axion dark matter. In Ref. [45], this work excludes dark photons in the frequency range from 0.5 to 5 Hz with a constraint of 1×10^{-5} to 1×10^{-6} on the kinetic mixing parameter.*”

Comment 1-4

Broadly speaking, the methodology is sound and there are enough details (if one includes the supplementary material) to reproduce these results.

However, there are a few points that I would like to raise:

For their correlations, they take data for 2000s at a sampling rate of 1000Hz ->so their single sided spectra should have 1M points.

However, their dark matter candidate bins are $\sim 320k$, which is much more than one would expect, and a significant fraction of the spectra (keep in mind this is at the 95% confidence limit!).

To me, this suggest that the “detection threshold” is (and finally their exclusion limits) being set too low; looking through the supplementary material, I think they are missing a statistical correction that should be dependent on the total number of frequency bins.

Our response: We thank the referee for pointing out this unclear description. At first, we would like to explain why the candidate bins are about 32% instead of 5% of the total bins. The detection threshold is determined through Monte Carlo simulation. In this simulation, we assume the absence of common-mode noise in the measurements, and consider the noises in the

15 magnetometers to be independent of each other. This is primarily due to the impracticality of incorporating such noise in simulated data. However, in actual measurements, common-mode noise exists due to Johnson noise originating from the mu-metal shields. Magnetometers within the same shielded room exhibit nearly identical phases of common-mode noise. Consequently, the SNR distribution obtained through cross-correlation is biased by the positive correlation between magnetometers within the same shielded room. This leads to a greater number of bins in the final SNR distribution surpassing the “detection threshold” established by the expected SNR distribution. To verify this, we have conducted a new Monte Carlo simulation that incorporates common-mode noise. Here the common-mode noise is set as about $5 \text{ fT/Hz}^{1/2}$. The resulting SNR distribution is obtained, and the number of exceeded bins can be calculated. Remarkably, this simulation yielded 318671 DPDM candidates, in agreement with the experimental outcome. Fortunately, these candidate bins can be eliminated through our exclusion methods, including the the comparison of long-baseline networks and “lagged frequency analysis”.

Considering this, we have added a new paragraph in Supplementary Information Section VIC to explain the source of these candidate bins as

“After conducting the initial analysis, we find that 318863 DPDM candidates exceed the threshold with 95% C.L., which is more than the stochastic fluctuation (around 50000). In the simulation (Fig. S7b), we assume that there is no common-mode noise and technical noise in the measurements. This is primarily due to the impracticality of incorporating such noise in simulated data. The common-mode noise is due to the Johnson noise from the mu-metal shields. Different magnetometers in the same shield room measure nearly the same phase of the common-mode noise. This implies that the SNR distribution obtained by cross-correlation is biased by the positive correlation between the two magnetometers in the same shield room. Hence, the final SNR distribution has more bins that exceed the “detection threshold” set by the expected SNR distribution.”

Comment 1-5

I find the idea of introducing a lag into one of the channels in order to suppress the DM signal and determine the background, clever.

If I understand correctly; they went through all their DM candidate bins and made sure they were consistent with the background using this technique.

Our response: We appreciate the referee’s approval of our “lag approach.” As the referee correctly recognized, we examined all 318863 DPDM candidate bins, concluding that all of them were consistent with background noise.

Comment 1-6

What worries me is that the limits presented in fig. 3b are the “detection threshold”, when the background is higher than the “detection threshold” (and therefore limits can not be placed at the threshold level)- is this the case?

ie. if a frequency bin has a detection threshold of SNR 2.5; and the measured bin SNR is 6; it will be flagged as a candidate.- after it is confirmed that the background corresponds to “SNR” of 6, is the limit set at 2.5 or 6?

This is particularly relevant given the large fraction/number of candidates.

Our response: We thank the referee for pointing out this unclear description. The limit should be set at “2.5” instead of “6”. Here the values “2.5” and “6” are provided as examples rather than realistic values in our experiments. However, for the sake of simplicity in our explanations, we will continue to use these values. Here we would like to explain the reason why we should choose “2.5” instead of “6”. The value “2.5” represents the detection threshold derived from requiring the integration area under the expected signal-to-noise ratio (SNR) distribution to be 0.95, which is determined through multiple measurements. This signifies that there is a 5% probability of exceeding the SNR of 2.5. On the other hand, the value “SNR=6” corresponds to the outcome obtained from a single measurement. It is reasonable to expect SNR values above the detection threshold (SNR=2.5) with a 5% probability, and these can be considered as candidates. After excluding these candidates, the SNR of 2.5 serves as the constraint for the 95% confidence level. This standard methodology is also widely employed in previous works, such as Searching for dark photon dark matter in LIGO O1 data [Communications Physics 2, 155 (2019)].

Comment 1-7

The authors ignore the width of the DM signal that would arise due to stochastic fluctuations (G.P. Centers et al, Nature Communications volume 12, Article number: 7321 (2021)) in their analysis.

The argument given is that the width (1/correlation time) at their maximum mass of 500 Hz matches their bin width of 0.5 mHz.

Nonetheless, scalloping and other windowing effects can lead to a significant reduction in the bin power (see for example Spectrum and spectral density estimation by the Discrete Fourier transform (DFT), including a comprehensive list of window functions and some new flat-top windows (2002) by G. Heinzel et al)

This effect would only be relevant for the higher frequencies in their limits. I would suggest either correcting the limits to include this or explicitly neglecting stochastic effects.

Our response: Indeed, this is a very important question and we thank the referee for this insightful suggestion. As the referee points out, the width of the DM signal indeed leads to a reduction of the bin power that we neglected in the previous version. After taking this effect into account, the correction on the DPDM constraints is not significant within our searching mass range and we have modified our constraints in the revised manuscript. Additionally, we have introduced a new section in the Supplementary Information titled “**Detection efficiency of dark photon**” to address this effect in detail. In the following, we would like to present the details of how we estimate the width effect on the DPDM constraints.

First, we consider the signal detection efficiency due to the signal power lost from binning, i.e., fractions of power falling into a single fixed bin. In a standard halo model, the velocity spread of local DPDM causes a frequency spread of $\Delta f/f_0 \approx 10^{-6}$, where f_0 is the Compton frequency of the DPDM. In detail, the normalized power spectrum of the DPDM signal is derived from the Maxwell velocity distribution, and the result is shown below in Figure a. For example, when $f_0 = 500$ Hz, the width of DPDM signal is $\Delta f = 0.5$ mHz. On the other hand, we acquire the experimental data with a measurement time T , where the bin size in the frequency domain is $1/T$. For example, the measurement time is $T = 2000$ s in our experiment and corresponds to the bin size of 0.5 mHz. We note that in this case, the DPDM signal power would experience power loss from the bin.

The power spectrum of the DPDM signal derived from the Maxwell velocity distribution is used to determine empirically the fractions of power falling into a single fixed bin, where bin boundaries are systematically varied over the allowed range. To do this, we perform a numerical simulation and the frequency f_0 of DPDM is uniformly distributed within the allowed range. The measurement time is set to be $T = 2000$ s, and we have simulated the detection efficiency for the DPDM signal range of 1-500 Hz. The resulting detection efficiencies (“Fraction of power detected”) with respect to the frequency are shown below in Figure b. As one can see, the detection efficiency is above 90% below 200 Hz and is about 75% at 500 Hz.

To provide a clear illustration of the modification, we have included a comparison between the updated limits and the previous limits in the figure below. As shown in the figure, the correction due to detection efficiency is not significant within the range of masses we investigated.

Considering this, we have updated the constraints in our study to incorporate this effect, and we have also modified Fig. 3b in the revised paper. Moreover, we have introduced a new section in the Supplementary Information to incorporate a discussion on this particular effect in our data analysis. The supplementary information is nearly the same with that presented above.

Comment 1-8

The authors also bring up the spatial dependence (similar to an TM mode) of the field that would be generated by the dark photons.

Maybe a way to further eliminate local common mode noise and boost sensitivity would be to combine the information of each magnetometer in the array, using their positions and orientation in this TM mode, before the correlation.

Our response: We really appreciate the referee for pointing out this valuable suggestion. Considering this, we have added several new sentences in the main text to introduce this strategy

on page 6, *“To enhance sensitivity even further, a promising strategy involves the installation of magnetometers at various positions or orientations within each shielded room. The information collected by these magnetometers can then be combined in an array to effectively reduce local common-mode noise (see Supplementary Information Section I).”*

Reply to Reviewer #2

Comment 2-1

The authors report new dark photon dark matter search limits 15 high-sensitivity magnetometers. The hunt for dark matter is one of the leading puzzles in elementary particle physics, and it has a profound transformative impact once discovered. It is extremely challenging and, hence, highly rewarding to make new progress in the field. Particularly impressive, the manuscript reports world-leading laboratory-based sensitivities in the 10-500 Hz regime and has a future projection that is possible to exceed the best astrophysical limits. The research is of high importance and high quality.

The 15 sensors are distributed in two sites, with 13 in Harbin and 2 in Suzhou. The cross-correlations between them would be a signature of the signal, which the authors use to suppress background.

I encourage the authors to address the following issues before I can consider recommending for publication:

Our response: We would like to thank the referee for his/her careful reading. The very insightful and valuable comments greatly help us to improve the quality of our paper. On the basis of these valuable comments, we have carefully modified our main text and Supplementary materials. To do this, we have added, for example, a detailed description of the atomic magnetometer, a discussion of advantages of the long-baseline network and a detailed explanation of the data analysis in the revised manuscript. In particular, we have modified Section VD to present the detailed exclusion procedure of DPDM candidates. In addition, we have added a new discussion to estimate the effects due to the Earth's rotation and angular difference between different network stations. In the following, we provide a detailed reply to the comments and the corresponding major changes in our revised paper.

Comment 2-2

General:

(1) In what sense these are quantum sensors? The devices used in this search are far from the standard quantum limit (which arguably have some sense of quantum). If not, the authors should change the name of "quantum sensor network" in the title and the text.

Our response: We understand the referee's concern and thank the referee for pointing out this comment. Quantum sensors utilize quantum systems, properties, or phenomena to measure physical quantities [C.L.Degen et al., Quantum sensing, Rev. Mod. Phys. 89, 035002 (2017)]. In the case of our atomic magnetometer, we employ the spin-exchange relaxation-free (SERF) effect, which is a quantum phenomenon. This type of atomic magnetometer is usually referred as SERF magnetometers and have been highlighted as an important type of quantum sensors in the review [C.L.Degen et al., Quantum sensing, Rev. Mod. Phys. 89, 035002 (2017)]. We would like to introduce more details what the quantum effects are employed in SERF magnetometers. Traditional atomic magnetometers suffer from relaxation processes due to collisions between atoms, limiting coherence time and measurement sensitivity. In contrast, SERF magnetometers operate with high-density alkali-metal vapor in small magnetic fields. In this regime, the rate of spin-exchange collisions surpasses the Larmor precession rate, causing atoms to precess only small angles between collisions. Previous works, such as ref. [I.K.

Kominis et al., Nature 422, 596 (2003)], demonstrated effective suppression of spin-exchange relaxation, enabling long coherence times in SERF magnetometers. Although current sensitivity does not reach the standard quantum limit, the utilization of quantum phenomena improves sensitivity to the femtotesla level. This sensitivity surpasses classical magnetometers like the fluxgate magnetometer and Gaussmeter (their sensitivities are usually at picotesla level).

Considering this, we have added the review “Quantum sensing” into our updated references list, and added several new sentences in the revised paper:

on page 3, “*We employ atomic magnetometers as DPDM detectors. Atomic magnetometers are recognized as a type of quantum sensor [C.L.Degen et al., Quantum sensing, Rev. Mod. Phys. 89, 035002 (2017)] that exploits the quantum phenomenon known as the spin-exchange relaxation-free (SERF) effect, thereby enhancing atomic coherence time and improving measurement sensitivity [42,43].*”

Comment 2-3

(2) What’s the gain of the long baseline? It seems the authors did not take much advantage of the long baseline, except that the two locations do not share some of the common systematics.

Our response: Indeed, this is a very important question. In the following, we would like to summarize the main advantages of long-baseline sensor network.

First, as mentioned by the referee, it is advantageous to mitigate common-mode noise through the utilization of a long-baseline sensor network. Currently, the primary source of common-mode noise stems from magnetic noise generated by shielded rooms. This specific type of common-mode noise can be effectively minimized by employing sensors placed in two separate shielded rooms. In such a scenario, the baseline between the two shielded rooms does not necessarily need to be excessively large. However, it is crucial to consider the potential impact of other large-scale noise sources on the experiment, such as the solar wind, geomagnetic storms, and Schumann waves. When the two shielded rooms are in close proximity, these types of large-scale noise may exhibit correlation across the sensors. Hence, it becomes imperative to separate the shielded rooms into two distant locations to mitigate the influence of such large-scale noise.

Second, although we assumed a standard halo model for the momentum distribution in the current work, other types of dark matter distributions, such as anisotropic ones like a local cold stream due to the merger of the galaxy, have been considered in literature. With long-baseline, our network has the potential to determine the nature of these types of dark matter and provide excellent localization.

Considering this, we have added new sentences to discuss on the large-scale noise as:

on page 4, “*Based on the results of numerical simulation, the dominant common-mode noise is due to the shield room. However, there exist large-scale noises potentially impacting on the experiment, such as the solar wind, geomagnetic storm, and Schumann resonances, and separating these two sites at a long distance can effectively reduce these noises.*”

Comment 2-4

Analysis:

There are several issues with the data analysis, assumptions, and processing:

(3) What is the purpose of using correlation treatments? It seems there would yield no difference by simply and classically adding all the 15 sensors and performing FT. Why not simply do the classical addition?

Our response: We would like to thank the referee for pointing out this unclear description. The correlation treatments applied in our experiment differ from classical direct addition. In the following, we would like to explain what their difference are and why we use correlation treatments instead of classical addition.

In experiment, we employ 15 atomic magnetometers as detectors for dark matter. It is worth noting that the sensitivities of these detectors differ from one another for each search frequency bin. In particular, some atomic magnetometers exhibit noise peaks that are independent from one another. We show that such noise peaks can be effectively suppressed beyond the capabilities of classical addition. To ascertain this, we conduct a numerical simulation. In this simulation, we generate 15 white noise spectra with equal amplitudes for the 15 atomic magnetometers. Subsequently, we introduce a significant noise peak in only one of the magnetometers. When classical addition is employed, the noise peak is reduced by a factor of approximately 15 due to the averaging effect with the white noise. In contrast, when correlation treatments are employed, the reduction ratio increases to approximately 100. This improvement originates from the fact that the noise peak remains uncorrelated among the magnetometers.

Moreover, we can generalize the aforementioned analysis to a general case and derive an explicit outcome. In a general setting, the sensitivities of N sensors are quantified by the variance of white noise, denoted as σ_m^2 where $m = 1, 2, \dots, N$. For the classical addition method, which involves the simple summation of data from N sensors and subsequent Fast Fourier Transform computation, the resulting variance can be expressed as $\sigma_{\text{classics}} = \sqrt{\sum_{m=1}^N \sigma_m^2} / N$. On the other hand, in the case of correlation treatments, the variance of the signal strength can be determined as $\sigma_{\text{correlation}} = \sqrt{\frac{\sum_{m=1}^N \omega_m \sigma_m^2}{N \sum_{m=1}^N \omega_m}}$, where $\omega_m = 1/P_m$ represents the weight assigned to each sensor and P_m denotes the power of the m # sensor. According to a straightforward mathematical calculation, we can prove that

$$\sigma_{\text{correlation}} = \sqrt{\frac{\sum_{i=1}^N \omega_i \sigma_i^2}{N \sum_{i=1}^N \omega_i}} \leq \sigma_{\text{classics}} = \sqrt{\sum_{i=1}^N \sigma_i^2} / N,$$

where the equality holds if and only if all σ_m^2 are equal. When the sensitivities are different, the correlation is better than classical addition. However, when the sensitivities are equal, the two methods yield identical results.

Considering this, we have added a new discussion in Supplementary Information Section IVA to explain the advantages of our correlation treatments. The supplementary information is nearly same with that presented above.

Comment 2-5

(4) Below Eq. S8, the authors assume a flat noise spectrum in adjacent 50 bins, is such assumption validated by test or other runs?

Our response: We appreciate the referee’s help in pointing out this question. We have already carefully examined the noise distribution across various frequency bins and have verified that the noise follows a flat Gaussian distribution. As an example, we analyzed the noise observed by the 10th sensor situated at Suzhou station. The noise data was collected with a center frequency of 325 Hz and a frequency width of 0.5 Hz. The plot below illustrates the distribution of the real part of the noise, which exhibits a well match with a Gaussian profile. Similarly, we have tested all other frequency bins and confirmed that the noise adheres to a flat noise distribution.

Considering this, we have added a figure in the Supplementary Information to show the distribution of noise. The figure is same as that presented above. Moreover, we have added new sentences in the revised Supplementary Information

below the Eq. S8, “*The assumption of flat noise is validated by the Gaussian test. The distribution of the real part of $z_{m,j}$ in a frequency window of 0.5 Hz is shown in Fig. S4. The distribution is Gaussian, which can show that the noise follows the same distribution as nearby frequency bins.*”

Comment 2-6

(5) The calculation of SNR and the corresponding signal limit setting process matters a lot for this works. In supplemental material B, the SNR calculation does show a sizable difference between experimental data and simulation in Fig.4, which does not support the statement “The SNR distribution within the window of 9.1-10.1 Hz is shown in Fig. S4b, which agrees well with that of the experimental data.” Similarly, the same sizable differences between the experiment and simulation are shown in Fig.5. These differences should have impacted the choice of SNR=3.49 which translates into 95% C.L. The authors should clarify their treatments.

Our response: We would like to thank the referee for pointing out this unclear description. We would like to explain the reason why the simulated SNR distribution is different from the experimental SNR distribution. In supplemental material B, the simulated SNR assumes the absence of common-mode noise and assumes the independence of noise across the 15 used magnetometers. However, practical measurements indeed reveal the presence of common-mode noise originating from magnetic disturbances within the mu-metal shielded rooms. Magnetometers housed within the same shielded room encounter common-phase noise, leading to a positive cor-

relation effect. Consequently, the SNR distribution derived through cross-correlation exhibits bias, resulting in a higher number of bins surpassing the “detection threshold” established by the expected SNR distribution. In order to validate this observation, we conducted a new Monte Carlo simulation which accounts for common-mode noise. The estimated value of common-mode noise is approximately $5 \text{ fT/Hz}^{1/2}$, calculated by assessing the Johnson current noise of the mu-metal shield room. As one can see, the resulting SNR distribution is in agreement with the experimental result. In our data analysis, we establish a detection threshold at the 95% confidence level based on the Monte Carlo simulation without common-mode noise. While this choice may yield more DPDM candidates, we can successfully exclude them through a candidate comparison conducted within long-baseline networks.

Considering this, we have added a new discussion in the Supplementary Information to clarify our data treatments. The supplementary information is nearly same with that presented above.

Comment 2-7

(6) In the more baseline and standard treatment, the authors found many excess signals. Then the authors used a lagged correction to eliminate false signals. From the description, it is unclear if the authors are lagging in the time domain (e.g., correlating signals taken at different times, differ by a large period, “we lag one of the signals (i.e., $z_{n,j}$) by a large period”) or by frequency (“We choose the frequency lag $\Delta j = 10$, which is larger than the intrinsic width of the DPDM signal.”). The authors should clarify.

Our response: We thank the referee for pointing out this unclear description. We performed the “lagged frequency analysis” in the frequency domain to check excess signals. In the lagged frequency analysis, the correlating signals are taken at different frequency bins. For example, the frequency bin for detector 1# used in the lagged calculation is f_j while the one for detector 2# is $f_j + \Delta j \times \Delta f$, where $\Delta f = 1/2000 \text{ Hz}$ is the frequency resolution.

Considering this, we have modified the sentence “we lag one of the signals (i.e., $z_{n,j}$) by a large period” to “we lag ... by a large frequency difference” in the revised paper. Moreover, we have added more details of the lagged frequency analysis process in the Supplementary Information (see also Comment 2-8 for details.)

Comment 2-8

(7) The above treatment, if implemented correctly can help reduce background (false signal) if they are from a correlated source, in contrast to the situation where backgrounds are of a thermal nature. The authors should comment on the motivation and rigor of the treatment. In particular, the authors used multiple lags whose purpose seems to solely suppress the background, is it legit (“Otherwise, the SNR will be computed again with the lag changed to $\Delta j + 10$ and repeat the above procedure.”)? Will it suppress the signal (also see next)?

Our response: We thank the referee for pointing out these unclear descriptions. First of all, we would like to explain the motivation of using lagged frequency analysis to exclude the false signal. In our experiment, there exist technical noise peaks impacting on our result and the peaks are broadband. By examining the correlation results using multiple lags, we can subtract the broadband correlated background noise in the measurement. Second, the using of multiple lags is to suppress the stochastic fluctuation within the broadband noise peaks. The false signal can go through one lagged frequency analysis due to stochastic effect and multiple lags can reduce this effect.

In the following, we would like to explain the rigor of our exclusion method. In our analysis, we assumed the velocity distribution of dark photon dark matter is the Maxwell distribution. The intrinsic width of the power spectrum of the signal $\sim 10^{-6} f_0$, where $f_0 = m'_\Delta / 2\pi$ is the initial frequency set by the dark photon mass. In other words, the dark photon dark matter signal spectrum is a narrow peak. Even for the maximum frequency $f = 500$ Hz, the intrinsic width is still within the frequency resolution $\Delta f = 0.5$ mHz. By choosing the $\Delta j = 10 \gg 1$ lagged bin, it is certain that the expected dark photon dark matter signal will not contribute to the lagged correlation and the multiple lags will not suppress the signal. For the sake of rigor, we performed a numerical test on the validity of our analysis strategy by injecting a fake signal. More details can be found in Comment 2-9. The lagged frequency analysis is useful to diagnose the broadband noise, however it is not capable to remove narrow band noise, whose width is within the frequency resolution. However, due to the signal of dark photon should have the same phase in all the sensors, the imaginary part of SNR must be zero. Hence, the narrow band noise can be suppressed by checking the imaginary part of SNR.

In order to clearly describe the lagged frequency analysis process, we have modified Section VD in the revised Supplementary Information as

“To ensure the validity of true DPDM signals and to effectively filter out false positives, it is imperative to identify the primary sources of noise and conduct a meticulous analysis of the DPDM spectral characteristics. In actual measurements, common-mode noise and technical noise are the dominant sources of the false candidates. The false signal due to common-mode noise is uncorrelated between the magnetometers in different shield rooms and can be eliminated by comparison of long-baseline networks. On the other hand, technical noise, characterized by correlated broadband noise with high SNR, may arise from factors such as sensor power supply instability. Notably, the frequency width of this broadband noise differs from that of true dark photon signals. In a standard halo model, the small but nonzero velocity spread of local dark matter causes a small frequency spread $\Delta f / f_0 \approx 10^{-6}$, where f_0 is the frequency of the dark photon. In our work, the searched frequency f_0 is below 500 Hz, and thus the intrinsic width of the dark photon signal Δf should be smaller than 0.5 mHz. Because the width of the DFT frequency bin is 0.5 mHz in our experiment, a true dark photon signal should be included inside a frequency bin. Based on the difference in frequency width, the lagged frequency analysis can be used to distinguish a true DPDM signal from noise. During calculating the cross correlation

for each two magnetometers, for example, $z_{m,j}$ and $z_{n,j}$, we lag one of the signals (i.e., $z_{n,j}$) by a large frequency difference, ensuring that no DPDM physical correlation between the two magnetometers should be expected.

In the lagged frequency analysis, a DFT frequency in one sensor is compared to a set of offset bins from other sensors such that a true dark photon signal would not contribute to a non-zero cross-correlation but for the broadband noise leads to a non-zero correlation. We choose the frequency lag $\Delta j = 10$, which is larger than the intrinsic width of the DPDM signal. For a potential signal bin j , the SNR ρ_j is the correlation between $z_{m,j}$ and $z_{n,j+\Delta j}$. For $C_{15}^2 = 105$ pairs of cross correlation, we implement the same lag procedure, and then calculate the corresponding SNR. When the SNR is still above the detection threshold, the potential DPDM candidate should be a false signal and can be excluded. Otherwise, the SNR will be computed again with the lag changed to $\Delta j + 10$ and repeat the above procedure. The using of multiple lags is to suppress the stochastic fluctuation within the broadband noise peaks. The broadband noise may go through one lagged frequency analysis due to stochastic effect and multiple lags can reduce this effect. After our exclusion procedure, the DPDM candidates obtained in Section VIC are all excluded.”

Comment 2-9

(8) In S-VI, the authors reconstructed the fake signal but lacked many details. Has the analysis gone through the “lag”-signal elimination process? If so, what is the SNR distribution after this “lag”-signal elimination process? It would also be more systematic that for the same experimental data set, the authors implement several signals at different frequencies and check whether the consistent signal reconstruction is true for different frequencies (so it is robust against background fluctuations).

Our response: We thank the referee for this question. We injected the fake signals at several different frequencies to test our data analysis procedure and the injected signals are visible in the average cross-correlation spectrum and SNR distribution. To check our exclusion procedure, we also checked the injected signals with “lagged frequency analysis”-signal elimination process. We find that the frequency bins with injected signals do not have any correlations with their neighbor bins, which is within expectation. Thus these injected signals pass the lagged frequency analysis and remain to be signal candidates. As we explained in SI-VI D, “When the SNR is still above the detection threshold, the potential DPDM candidate should be a false signal and can be excluded.” In this sense, the injected signal is a real DPDM signal, and the other peaks should be false signals and can be excluded. Considering this, we have added a new paragraph in SI-VII to discuss about the signal elimination process.

Comment 2-10

Others:

(9) (Minor) Given that we do not know the DM direction, we are actually constraining the component of DM along the vertical direction of the experiment. There are a few effects, including the rotation of the direction over 2000 sec of data collecting time, the angular difference between site A and site B, etc. It might be useful to comment on their impact (which is expected to be small but gives an $O(1)$ suppression to the signal rate).

Our response: We concur with the referee’s observation that our experimental configurations measure the dark photon wavefunctions projected along the vertical axis. As a result, certain

factors, such as the Earth’s rotation over the entire measurement period and the angular separation between the Harbin and Suzhou sites, may have a minor impact on the dark photon-dark matter signals. In the following, we will assess the extent of this influence on the DPDM signals.

The two sites, Harbin and Suzhou, are separated by approximately 1400 km. The angular difference along the vertical axis between these locations is roughly $1400/R_E \approx 0.22$, where $R_E \approx 6371$ km denotes the Earth’s radius. This angular disparity translates into a signal suppression factor of $\cos(1400/R_E) \approx 0.98$ for cross-correlations between magnetometers located in different shield rooms. Similarly, we estimate the impact of Earth’s rotation over a 2000-second interval. This rotation induces a sideband with a frequency of 1 day^{-1} , distinct from the dark photon-dark matter peak frequency. However, this sideband is not expected to be resolved due to the frequency-domain resolution of $1/2000 \text{ s}^{-1}$. For the influence on signal amplitude, a detailed analysis necessitates modeling the dark photon-dark matter wavefunctions across the entire time series. We provide a conservative estimate for the suppression factors: $\cos(2000 \text{ s} / 1 \text{ day}) \approx 0.9997$, which is negligible.

In response to this feedback, we have incorporated a concise discussion of these effects in the Supplementary Information of the revised version. The supplementary information is nearly the same with that presented above.

Comment 2-11

(10) (Minor) Fig.2 should only have two columns and rows for site B, for the configuration realized in this work.

Our response: We thank the referee for pointing out this. We have changed the numbers of columns and rows for site B in Fig. 2. Once again, we thank the referee for his/her insightful and valuable comments and suggestions.

Reviewer #1 (Remarks to the Author):

Overall, I think the authors have adequately addressed most of the points I have raised before.

- They have toned down the general claim to originality.
- They included the very relevant work by Sulai et. al. And also put their limits in their figures.
- They responded to my concerns regarding the large number of candidates & their setting of the threshold.- (more on this below)
- They included a satisfactory correction for the DM-decoherence effect. This was done by scaling their limits by the fraction of power expected to be integrated in a single frequency bin [Similar technique to other, ie. Tretiak PRL 129, 031301, detailed in the Supp. Mat. section 'Decoherence losses', maybe this could be cited].
- They are missing a correction of the look-elsewhere effect. Is it not necessary?
- They introduced the idea of using the geometry of the TM-like mode excitation for further noise rejection.

###

Regarding the large fraction of candidates & setting 'low' limits after confirming the background level.-

As a first reaction, I am skeptical of a DM search that flags 1/3 of measurements as DM candidates. And proceed to 'eliminate' all of them by confirming they have 'excess' noise in 1/3 of their points. Then, the obvious questions are: could they have eliminated a real DM detection? how good is the noise characterization? can they reject the noise well enough to still place limits at the claimed level?

However, I would not gatekeep this work from getting published on these grounds, as:

- 1.- The authors have clearly thought this through and can defend their work.
- 2.- The authors are correct in pointing out what they have done follows common procedure and meets the expected standards in the field.
- 3.- They are transparent about the large amount of candidates. This should be further improved by clearly indicating in the main text what % of the total frequency bins this is: ie. the phrase "Our analysis procedure yields 318863 potential DPDM candidates that exceed the 95% C.L." should be changed to: "Our analysis procedure yields 318863 potential DPDM candidates (~XX% of total frequency bins) that exceed the 95% C.L."
- 4.- Their exclusion limits clearly indicate limited sensitivity in noisier frequencies.

###

For all these reasons, I would the say the work is generally OK and support publication with the minor revisions suggested above.

Reviewer #2 (Remarks to the Author):

The authors have addressed the issues raised in the previous report in a satisfying way. I have no further questions about this manuscript and I recommend it for publication at Nature Communications.

Reply to Reviewer #1

Comment 1-1

They are missing a correction of the look-elsewhere effect. Is it not necessary?

Our response: We understand the referee's concern and appreciate the opportunity to address it. It is important to note that in our experiment, the correction for the look-elsewhere effect is deemed unnecessary. When searching for signals, there exists the possibility of identifying something seemingly significant that may actually stem from random fluctuations in the noise. This likelihood of observing a signal by sheer chance is heightened when multiple independent statistical tests are conducted. However, instead of discovering DM signals from noise, we set the constraints after eliminating the false DM candidate. In our specific case, we eliminated all the false DM signals and calculated the threshold of Gaussian noise under 95% confidence level. Consequently, there is no need to include the correction of look-elsewhere effect.

Comment 1-2

As a first reaction, I am skeptical of a DM search that flags 1/3 of measurements as DM candidates. And proceed to 'eliminate' all of them by confirming they have 'excess' noise in 1/3 of their points. Then, the obvious questions are: could they have eliminated a real DM detection? how good is the noise characterization? can they reject the noise well enough to still place limits at the claimed level?

Our response: We understand the referee's concerns and would like to thank the referee for pointing out this comment. First, we would like to emphasize that our elimination procedure will not eliminate a real DM detection. The real DM signal should be correlated between any pairs of detector and will not show up in the lagged correlations. Additionally, in our data analysis procedure, we injected several fake DM signals at different frequencies and checked the injected signals with "lagged frequency analysis". We find that the frequency bins with injected signals do not have any correlations with their neighbor bins, which is within expectation. Thus these injected signals pass the lagged frequency analysis and remain to be signal candidate. Second, we characterized the noise well by numerical simulation before elimination procedure. In the simulation, $5 \text{ fT/Hz}^{1/2}$ common-mode noise (calculated by assessing the Johnson current noise of the mu-metal shield room) was added into 15 independent Gaussian noise and the resulting SNR distribution is in agreement with the experimental result. At last, common-mode noise and technical noise are the dominant sources of the false candidates. The false signal due to common-mode noise is uncorrelated between the magnetometers in different shield rooms and can be eliminated by comparison of long-baseline networks. On the other hand, technical noise, characterized by correlated broadband noise with high SNR, may arise from factors such as sensor power supply instability. Based on the difference in frequency width, the lagged frequency analysis can be used to distinguish a true DM signal from noise.